# Mental Strength and Challenges among Thai Medical Students in Their Clinical Years—Study Protocol

**DOI:** 10.3390/healthcare9030305

**Published:** 2021-03-10

**Authors:** Tanrin Hiranwong, Patipan Sitthiprawiat, Sirinut Siritikul, Jiraphat Jiwtrakul, Sirilux Klaychaiya, Pookit Chaipinchana, Pimolpun Kuntawong, Tinakon Wongpakaran, Nahathai Wongpakaran, Athavudh Deesomchok, Danny Wedding

**Affiliations:** 1Faculty of Medicine, Chiang Mai University, 110 Intawaroros Rd., T. Sriphum, A. Muang, Chiang Mai 50200, Thailand; tanrin_h@cmu.ac.th (T.H.); patipan_s@cmu.ac.th (P.S.); sirinut_siritikul@cmu.ac.th (S.S.); jiraphat_j@cmu.ac.th (J.J.); sirilux_klaychaiya@cmu.ac.th (S.K.); pookitpun@gmail.com (P.C.); athavudh.d@cmu.ac.th (A.D.); 2Department of Psychiatry, Faculty of Medicine, Chiang Mai University, 110 Intawaroros Rd., T. Sriphum, A. Muang, Chiang Mai 50200, Thailand; pimolpun_k@cmu.ac.th (P.K.); nahathai.wongpakaran@cmu.ac.th (N.W.); 3Saybrook University, Pasadena, CA 91103, USA; danny.wedding@gmail.com

**Keywords:** well-being, mistreatment, mental health, character strength, medical students, clinical

## Abstract

(1) Background: Mental well-being and mental health problems are both important, especially among medical students who will be future doctors. The proposed study aimed to explore both positive and negative mental health experiences, especially mistreatment, occurring among medical students in their clinical years. (2) Methods/design: The study will conduct a cross-sectional survey between January 2021 and December 2021, among medical students studying in their clinical years across 23 medical schools throughout Thailand. Measurements regarding character strengths related to medical professionalism as well as other positive mental health strengths and negative mental health problems, e.g., anxiety, depression and experience of mistreatment will be completed. Both medical students and faculty members will be invited to participate in the study. (3) Discussion: this survey will provide an overall picture of medical students’ mental well-being, positive and negative aspects of mental health and the magnitude of mistreatment and perspectives they experience. The limitations of the survey will be discussed.

## 1. Introduction

Ensuring medical students’ psychological, social and physical well-being is an ongoing concern for any medical school, and in many ways these factors are as important as academic achievement. A systemic review revealed that the estimated prevalence of depression or depressive symptoms among medical students reached 27.2%, and suicidal ideation was 11.1% [1,2]. In Thailand, another study showed that 30% of medical students were clinically depressed, while 12% had suicidal ideation [3]. We intended to develop a program aimed to optimize not only academic achievement but also professionalism and student well-being [4,5]. Many factors are related to medical students’ academic achievement, mental health problems and well-being. At the individual level, personal strengths and perceived social supports played important roles in motivation for studying, mental health and well-being [6,7].

Well-being is an elusive term, rooted in the concepts of hedonia (pleasure-seeking) and Eudaimonia (a contented state of being happy, healthy and prosperous). Hedonia consists of two components: life satisfaction and the balance between positive and negative affects [8,9], while eudaimonia describes a broad type of well-being. When investigating the measurements using the two concepts, no discriminant validity has been found. However, researchers are encouraged to evaluate and investigate specific variables, e.g., character strengths, rather than hedonia and eudaimonia [10]. Some medical educators have adopted Dodge et al.’s recommendation in defining well-being, quoted as, “when individuals have the psychological, social and physical resources they need to meet a particular psychological, social and/or physical challenge” [11]. For example, when individuals have sufficient resilience, they can cope with stress, symptoms of anxiety or depression and burnout [4]. Well-being is also proposed in six components which are purpose in life, environmental mastery, positive relationships, personal growth, autonomy and self-acceptance [12]. Although factors that affect the well-being may vary according to cultural influences, personality or social background [8,13], the six core values remain the same. Well-being is not a stable state. Studies have strived to find ways to achieve a sustainable change of happiness or well-being. Diener, Lyubomirsky and colleagues [8,14] found that happiness or well-being is regulated by genetics influencing the set point of well-being, e.g., personality trait and intentional activity, practicing positive “virtues”, e.g., gratitude and happiness-relevant circumstantial factors, e.g., childhood trauma.

Medical students may have different ways of developing and sustaining their sense of well-being compared with people in general. Being accepted as a medical student may be the first time one feels a boost to their sense of well-being. Medical students not only need to succeed academically but need to learn and internalize positive psychological attributes to become a competent doctor. Another time that boosts their sense of well-being is when they encounter a real patient in the 4th year. However, becoming a junior student surrounded by senior students and experienced doctors could result in emotional stress. Studies showed that medical students’ well-being was associated not only with their perceptions of supportive learning environments but also with empathy, moral reasoning and tolerance of uncertainty that they possess [15].

Related research supports that medical students experience higher stress levels, higher rates of burnout, poorer quality of life, depressive symptoms and suicidal ideation more than young adults [16,17].

Character strengths are related to academic achievement, mental health problems and well-being. They are related to medical professionalism including self-regulation, an ability to control one’s emotions [18]; gratitude, a thankful feeling and/or appreciation [19]; prudence, thoughtful, logical thinking, and the ability to plan methods to achieve goals [18]; humility, defined as a state of mind that is humble or free from arrogance [18]; and resilience, referring to a process of adapting well when facing stressors such as trauma, tragedy or threats. Employing these character strengths increases happiness [20] and enhances positive feelings resulting in a greater overall sense of well-being [19]. Further, these characteristics enhance positive relationships with patients, improve clinical outcomes [19] and are associated with well-being, stress coping, low rate of burnout, a reduced sense of victimization and improvements in medical students’ mental health [21,22,23,24,25]. Moreover, feeling supported by family members, friends or significant others helps to reduce stress and mental problems and enhances positive mental health and self-esteem [26].

Apart from the individual level, the institutional level is also important. Mental health and well-being of medical students can be supported by a variety of methods such as extracurricular activities that promote character strength and well-being. Medical schools can provide supportive environments that promote students’ personal growth while difficulties or distress can be safely addressed and managed to improve medical students’ mental well-being [27].

Despite these efforts, medical students still experience mental health difficulties. Relationships with peers and colleagues play an important role and directly affect medical students’ psychological state. Study in the clinical years (four to six) inevitably involves difficult encounters with numerous people including residents and senior faculty. These encounters allow students the opportunity to acquire knowledge and develop the attributes of a competent physician; on the other hand, they can also lead to the potential for being mistreated. Mistreatment is defined as “either intentional or unintentional acts occurring when behavior shows disrespect for the dignity of others and unreasonably interferes with the learning process.” Mistreatment can include sexual harassment; discrimination or harassment based on race, religion, ethnicity, sex, or sexual orientation; humiliation; psychological or physical punishment; and the use of grading and other forms of assessment in a punitive manner [28]. Mistreatment has been reported among 4 to 41% of students across medical schools. The most common form of mistreatment was public humiliation from clinical professors [28].

Problems like these can arise in any society but are somewhat less likely in a highly structured environment such as a medical school. However, being mistreated can affect academic performance [29,30], and some students respond to mistreatment with stress, burnout, anxiety, depression and decreased motivation to continue studying [31,32,33,34]. Other problems include the inability to provide high quality of care to patients, personal shame and doubt, relationship problems or substance misuse and suicide [33,35,36] (Figure 1). To what extent mistreatment affects medical students’ quality of life remains unknown.

Mistreatment can be expressed in a variety of forms, and societal and cultural factors may be involved. Some mistreatment behaviors may be viewed or interpreted differently by other cultures and those who are involved with mistreatment [37,38]. Sadly, the number of reports of mistreatment among medical students seems to be growing across cultures [31,32,34].

Although mistreatment remains a challenging issue in medical education, problems are preventable with appropriate communication and a clear understanding between medical personnel and students [33,35,39]. However, several studies have shown that managing systems to eliminate such problems remains ineffective and unsafe [28,32]. In contrast, cultivating positive psychology attributes can reduce rates of depression and increase positive aspects of medical school and cultivate a sense of well-being [40,41,42]. Whether positive virtues or character strengths can buffer or prevent the negative psychological consequences after mistreatment remains unknown.

### Current Study

This study proposed to promote medical students’ academic performance, mental health and quality of life. We aimed to determine those factors that either positively or negatively affect medical students. The study aimed to explore character strengths among medical students, especially those related to medical professionalism, as well as any mistreatment experiences, all of which are related to their psychological well-being. In addition, students’ mental health status and quality of life were explored. Because faculty members may be involved in medical student mistreatment, investigating this issue was crucial [31,37,38,43]. In this study, the authors asked instructors and faculty members to participate to examine their perspectives on mistreatment as well as their own mental health and quality of life.

Based on aforementioned studies, we hypothesized that medical students would experience mental health problems, e.g., perceived stress, depression and anxiety the same as nonmedical undergraduate students in the university. We expected the medical student to have high levels of character strength, motivation, perceived social support, psychological well-being and quality of life. We expected to discover some mistreatment incidence and hoped to obtain some useful information related to that incidence, e.g., supporting system. We hypothesized that students’ strength would reduce or become a buffer against stress and psychological problems.

## 2. Materials and Methods

### 2.1. Study Design and Time-Period

The study will employ a cross-sectional design among medical students and faculty members currently studying and working in 23 qualified medical schools throughout Thailand (12 December 2019).

### 2.2. Study Population

The participants will be medical students currently studying in their clinical years, years 4 to 6. Inclusion criteria include students having (1) an electronic device with an internet connection such as mobile phone, tablet or personal computer to submit questionnaires and (2) they will have passed at least one rotation in clinical training. For instructors, the criteria include (1) currently working within one of the 23 certified Thai medical schools, (2) having at least 1 year of teaching experience in clinical training and (3) possessing an electronic device with internet connection such as mobile phone, tablet or personal computer to submit questionnaires. 

### 2.3. Procedure and Participant Invitation

Because of COVID-19 pandemic requirements for physical distancing, we developed an online questionnaire for this study. The investigator team will provide the relevant link or the Quick Response code (QR code) to all potential participants. Flyers to invite students to participate in the study will be placed in private areas such as medical students’ dormitories or private rooms for medical students in teaching hospitals. Social media networks such as LINE and medical student associations will be used to communicate and distribute the questionnaires. A convenience and snowball sampling strategy will be applied to recruit potential participants. No reimbursement, gifts or payments will be offered to compensate for completing the questionnaires.

Data will be collected for one year. Information regarding informed consent and details about the study will be included on the first page of the questionnaire before participants answer questions to ensure that participants understand and willingly participate in the study. Personal details that could be used to identify respondents will be excluded from the questionnaire. More details can be found in Table 1.

### 2.4. Measurements

All individuals will complete the surveys on their own. The primary questionnaires will include sociodemographic data, information related to participants’ status, information related to support and extracurricular activities provided by the faculty, followed by questionnaires assessing positive qualities including the Resilient Inventory (RI-9) measuring the extent to which individuals feel confident to overcome difficulties in life [44]. In addition, PhuSeG scale, a 10-item composite scale assessing prudence, humility, self-regulation and gratitude [45], Thai version Rosenberg Self-Esteem Scale (RSES) measuring global self-worth including both positive and negative feelings about the self [46] and the Revised Thai Multidimensional Scale of Perceived Social Support (rMSPSS), measuring the extent to which an individual feels support by family members, friends and significant others will also be used [47,48]. The Academic Motivation Scale (AMS) assessing three types of motivation based on the self-determination theory, i.e., intrinsic, extrinsic and amotivation [49], Thai version of the Perceived Stress Scale (T-PSS-10) measuring the extent to which individuals perceived stress [50] and EQ-5D used for quality of life scoring [51] will also be included. In addition, a visual analog of Burnout scale, a single-item measure of burnout ranging from 0 to 100, Core Symptom Index (CSI-15) designed to measure anxiety, depression and somatization symptoms [52] and Traumatic Experience Scale (TES) addressing potentially traumatizing events will be employed. The items include emotional neglect, emotional abuse and physical abuse from family members, and they were developed by T. Wongpakaran and N. Wongpakaran. The instrument uses a 5-point Likert scale, ranging from 1 (never) to 5 (almost always). The score ranges from 25 to 125, the higher the score the higher the level of traumatic experience. Preliminary findings suggested that the TES is a reliable and valid self-report instrument that can be used in clinical practice and research (Cronbach α = 0.91). Lastly, Mistreatment Questionnaires will investigate the participants’ experiencing or witnessing an episode of mistreatment. More details can be found in Table 2.

### 2.5. Statistical Analysis Plan

#### 2.5.1. Sample Size Calculation

The minimal numbers of required participants for the cross-sectional study was calculated using a formula developed by Jaykaran Charan and Tamoghna Biswas [53]. A related study conducted at a Thai medical school in the southern region [31] indicated that the proportion of medical students experiencing mistreatment at least once during their clinical years totaled 63.4%. With the absolute error of 5% (d = 0.05) and type I error at 5% (Z = 1.96), we calculated the numbers of required participants; these calculations indicated we needed to recruit at least 357 participants for this study. As related research has not provided estimates of the proportion of medical professors mistreating medical students compared with all medical professors, we assume that the proportion is 0.5; therefore, we will need to recruit at least 385 medical professors to participate in this study. However, because we are using an online survey method, we do not anticipate any difficulty recruiting subjects. In this study, the participants do not receive any remuneration.

#### 2.5.2. Statistical Analysis

Descriptive statistics will be used for sociodemographic data presented as frequency; percentage (%) will be used for categorical variables, e.g., sex, while continuous variables, e.g., age and resilience inventory score, will be presented as mean ± standard deviation or median (min–max). To assess differences between groups, chi-square, *t*-test, and ANOVA will be used. Correlation and regression will be used to examine associations between anticipated outcomes and predictors. *p*-values < 0.05 will be considered significant correlations. STATA, Version 13.0, and SPSS, 22.0 will be used for data cleaning and statistical analysis. Multiple imputations will be used for missing data or incomplete responses.

### 2.6. Ethics Approval, Consent to Participate, Autonomy and Confidentiality

The study has been approved by the Ethics Committee of the Faculty of Medicine, Chiang Mai University. The invitation process will be conducted without inducement or coercion. Students who volunteer to participate in this study will submit anonymous self-reported questionnaires, and the research will be entitled, “Educational experience in clinical years in Thai medical schools” to avoid leading questions that result in biased responses. Individual identification will not be required to ensure participants’ anonymity and safety. Neither participant identification data nor university name will be mentioned in any report or publication. Communication between researchers and participants will be conducted by assigned representatives in each university using contact numbers and the email address of the researchers. A written informed consent embedded on the first page of the online questionnaire will have to be signed electronically before anyone completes the questionnaires. When participants respond ‘No” to this question, it will be deemed a refusal to participate, and the process will be ended. At the conclusion of each questionnaire, advice regarding safety issues, e.g., coping with suicidal ideation, will be provided, and students will be told where they can go and what they can do to seek help. All participants can leave any item(s) unanswered if they feel uncomfortable responding to the item(s).

## 3. Discussion

This study constitutes the first nationwide evaluation of mental health, character strength, mental health problems, mistreatment and quality of life among medical students completing their clinical years in Thailand. Assessing the mental health of medical students is important to ascertain how well they are prepared to enter a highly demanding profession like medicine. Psychological well-being is important for medical students, and they also need to learn how to maintain this state. In addition, medical students have to learn to balance competing needs. For example, on the one hand, medical students need to develop interpersonal sensitivity and be attuned to patient’s emotions [54]; on the other hand, they may easily burnout as a result of such sensitivity [55]. As noted earlier, psychological well-being can be temporary according to the concept of the hedonic treadmill which suggests that any gains in happiness or well-being are only temporary, because humans so quickly adapt to change. Based on Lyubomirsky et al., to create sustainable well-being, students may need to work on promoting intentional or activity-related character strengths such as gratitude, resilience and self-regulation. This could promote the enjoyment of positive life experiences so that life satisfaction and enjoyment are extracted from one’s circumstances such as childhood trauma [14]. At the same time, negative affects derived from the past childhood trauma and current mistreatment need to be tackled. Well-being is not unitary but involves multidimensional aspects and concepts also have a number of implications [8]. This study will provide information regarding the potential strengths and weaknesses of the students we investigate, which would be useful for any further interventions.

The study will also reveal psychological features that could be related to variables such as self-esteem or motivation for studying medicine. Study findings also can be compared with those of medical students in other countries. The same is true with some character strengths, perceived stress and perceived social support, all of which have been shown to be related to student well-being and clinical performance.

We intend to ask participants about their childhood traumatic experiences. This information will provide insights about vulnerability to stress or mistreatment. More importantly, we are curious to determine whether early traumatic experiences would be related to students’ mental health difficulties and well-being. One study showed the direct relationship between childhood trauma and school bullying victimization [56]. However, evidence on the correlation between childhood trauma and mistreatment, regardless of being a person who is mistreated or who mistreats, is scarce. When a relationship is identified, it may be possible to identify and implement an intervention to prevent or reduce the influence of the history of mistreatment.

Another important issue related to childhood traumatic experience is unstable personality traits [57]. Recent findings have shown that university students have a high prevalence of unstable personality conditions such as borderline personality disorder [58,59]. Such an unstable personality could easily lead to mental health problems and affect psychological well-being. Only a few studies have investigated this important issue among medical students [60]. Our study will provide an opportunity to explore these relationships. For example, we will be interested in examining whether resilience influences the link between mistreatment and anxiety or depressive symptoms. We also hope to understand whether perceived social support plays a role in increasing self-esteem when students experience mistreatment.

In addition, the study will provide us insights concerning how well the students perceive happiness-relevant activities provided by the faculty through extracurricular activities [14], particularly, how much access and help they receive in times of need, e.g., when they are depressed or mistreated.

Some limitations of the proposed research are, firstly, the study employs a cross-sectional design, and this will preclude any examination of causal relationships. However, this initial study will set the stage for further and deeper studies in the future. Secondly, personality traits that are related to both negative mental health problems and psychological well-being [14] will not be included in this study.

## 4. Conclusions

This cross-sectional study will provide a snapshot of strengths and weaknesses, both at individual and institutional levels. It will alert those involved in the process to recognize the well-being medical students possess, how much difficulty they experience psychologically, the extent and magnitude of the ongoing problem of mistreatment and what factors are involved. All these data will provide insights to help us clearly plan the next steps in enhancing medical students’ well-being in Thailand.

## Figures and Tables

**Figure 1 healthcare-09-00305-f001:**
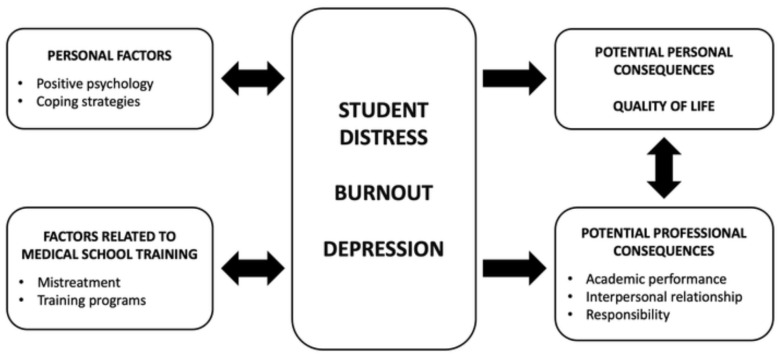
Relationship between student distress, personal factors, potential personal consequences and factors related to medical school training.

**Table 1 healthcare-09-00305-t001:** Sociodemographic information.

Demographic	Choices
Sex	Male
	Female
	Other
Sexual orientation	Homosexual
	Heterosexual
	Bisexual
	Others
Age	22 to 24 years old
Physical disease	Hypertension
	Diabetes
	Hyperlipidemia
	Hyperthyroidism
	Asthma
	Others
	None
Mental health problem	Depressive disorder
	Persistent depressive disorder
	Anxiety Disorder
	Obsessive Compulsive Disorder
	Bipolar disorder
	Others
Religion	Buddhism
	Christianity
	Islam
	None
	Others
Current habitat	House
	Faculty dormitory
	General dormitory
	Renting house
	Others
Whom do you live with?	Family member
	Roommate
	No one
Monthly income (THB)	Less than 5000
	5000–10,000
	10,000–15,000
	More than 15,000
Ward/Department on your rotation at the present	e.g., Medicine, Surgery, Pediatrics, Obstetric and
	Gynecology
Ward/Department you have been to	e.g., Medicine, Surgery, Pediatrics, Obstetric and
	Gynecology
Grade Point Average (GPA)	

**Table 2 healthcare-09-00305-t002:** Measurement tools of the survey.

Instrument	Aim in Assessing	Response Format	Number of Items	Recall Period	Internal Consistency
*Resilient Inventory* (RI-9)	Level of resiliency	5	9	Current	0.88
*PhuSeG scale*	Level of character strength	5	10	Current	0.89
*Thai version Rosenberg Self-Esteem scale (RSES)*	Level of self-esteem	4	10	Current	0.86
*Revised Thai Multidimensional Scale of Perceived Social Support (rMSPSS)*	Level of perceived social support	7	12	Current	0.91
*Academic Motivation Scale* (AMS)	Level of motivation and amotivation	7	28	Past to present	0.84
*Thai version of Perceived stress scale (T-PSS-10)*	Level of perceived stress	5	10	Past 4 weeks	0.85
*EQ-5D*	Quality of life	5	5	Current	0.87
*A visual analog of Burnout scale*	Level of burnout	10	1	Current	1.00 (CVI)
*Core Symptom Index (CSI-15)*	Severity of psychological distress	5	15	Past 1 week	0.85
*Traumatic Experience Scale (TES)*	Level of childhood traumatic experience	5	25	Childhood	0.91
*Composite Questionnaire of Mistreatment*	Qualitative and quantitative data of mistreatment experience	mixed	13	Past to present	1.00 (CVI)
*Scenario regarding mistreatment*	Perspective on the situations may be related to mistreatment	4	11	Current	1.00 (CVI)

Internal consistency is calculated using Cronbach’s alpha coefficient (>0.7 considered acceptable); CVI = content validity index (>0.8 considered acceptable).

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
