# Peer review of "Mental Strength and Challenges among Thai Medical Students in Their Clinical Years—Study Protocol"

_healthcare, 2021, doi:10.3390/healthcare9030305_

Round 1
Reviewer 1 Report
The topic is interesting and novel. But they haven't done the research.
I advise you to do the research first and when you have the results, send them to consider for publication. Journals do not usually publish research proposals.
Author Response
Dear reviewer,
We would like to thank you for your comments. Before submitting the manuscript, we have checked the journal's scope and found that the journal has accepted the study protocol. However, we have revised our manuscript based on another reviewer’s comments.
Reviewer 2 Report
Great idea for a proposed study. Good intro and proposed list of measures. Great use of tables: clear and succinct, easy to follow.
A few recommendations:
1. Keep this study focused on medical students' mistreatment and/or experiences in medical school, and screen out participants with histories of trauma, childhood troubles, and/or personality struggles such as BPD. These issues are extremely complicated and have profound, far-reaching effects on people's interpretation of social experiences; they would easily confound the results of a study about experiences in medical school.
2. Then, a separate study could be done examining the effects of childhood challenges, traumatic experiences, and/or personality struggles on performance in medical school.
3. The proposal currently seeks to explore multiple aspects of medical students' experiences: character strengths, professionalism, mental health, and quality of life. The broad nature of these variables make the proposal lean more toward a qualitative study. Perhaps consider reducing the dependent variable to just one of the above. Or change the study to be a qualitative examination.
Author Response
Dear Reviewer,
We would like to thank you for your comments. We totally agree that the project has included many aspects, both positive and negative. However, this is our purpose of the research project to see how positive attributes could be a buffer against negative outcomes such as stress. We have received approval from the Ethics committee and received funding from the Faculty of Medicine, Chiang Mai University to conduct this study. However, once we obtain the data, we plan to make the best use of the data publishing an article based on the particular outcome to meet the best interests of the readers. For this study protocol, we have to keep this manuscript the same. In fact, we have revised our manuscript based on another reviewer’s comments. Thank you very much.
Reviewer 3 Report
Dear Editor,
please find enclosed my comments on the manuscript “Mental strength and challenges among Thai medical students in their clinical years – study protocol", which was submitted to the Healthcare. My general impression after reading this manuscript is that this article is suitable, when it comes to the content and format, for publication in the Healthcare. However, it needs some major revision, especially when it comes to better/more clear description of particular parts of this manuscript. Below I summarized my comments in respect to the specific parts of this manuscript:
Introduction – the authors underscore the issue of well-being, however, the introduction fails in presenting this issue, especially the classic theoretical perspectives on well-being (see e.g. hedonistic perspective by Diener vs. eudemonistic perspective by Ryff). Moreover, in the introduction there should be more discussed the debate on the process of the adaptation of subjective well-being (SWB) as result of experiencing various stressful life events (see stress faced by the students, including ordinary daily stress, but also traumatic stress). The authors should indicate to what extent well-being is stable (see hedonic treadmill” model; Lyubomirsky, 2011), and also in-born, personality oriented feature (see the role of personality traits in well-being and its change) or is changeable and depend from the current life-situations and stressors. What is the difference in adaptation of well-being in general versus clinical samples. Thus, please discuss and cite following papers on that issue:
Diener, E., Heintzelman, S. J., Kushlev, K., Tay, L., Wirtz, D., Lutes, L. D., & Oishi, S. (2016). Findings All Psychologists Should Know From the New Science on Subjective Well-Being. Canadian Psychology/Psychologie Canadienne, doi:10.1037/cap0000063
Lyubomirsky, S., Sheldon, K. M., & Schkade, D. (2005). Pursuing happiness: The architecture of sustainable change. Review of General Psychology, 9, 111–131.
Ryff, C. (2014). Psychological Well-Being Revisited: Advances in the Science and Practice ofnEudaimonia. Psychotherapy and Psychosomatics, 83, 10–28. doi: 10.1159/000353263
Rzeszutek, M., Gruszczyńska, E., Firląg-Burkacka, E. (2018) Socio-medical and personality correlates of psychological well-being among people living with HIV: a latent profile analysis. Applied Research in Quality of Life, 14, 1113-1127. doi: 10.1007/s11482-018-9640-1.1037/1089-2680.9.2.111
Finally, at the end of introduction, before the method section please provide current study section, which you highlight aims of the study and hypotheses.
Method – this part is described well, but please put the table with sociodemographic information of the sample. Did participants receive some renumeration for study participation?
Results – this part is very well presented.
Discussion – this part, like introduction, is poorly grounded in the classical literature on well-being. Thus, I recommend also to elaborate on this issue more explaining the results they obtained - above papers to cite and discuss also here will be helpful.
Taking everything into an account, I recommend publishing this manuscript in the Healthcare. However, it needs before some major revisions, which I highlighted above.
Author Response
Dear Reviewer,
please find enclosed my comments on the manuscript “Mental strength and challenges among Thai medical students in their clinical years – study protocol", which was submitted to the Healthcare. My general impression after reading this manuscript is that this article is suitable, when it comes to the content and format, for publication in the Healthcare. However, it needs some major revision, especially when it comes to better/more clear description of particular parts of this manuscript. Below I summarized my comments in respect to the specific parts of this manuscript:
Introduction – the authors underscore the issue of well-being, however, the introduction fails in presenting this issue, especially the classic theoretical perspectives on well-being (see e.g. hedonistic perspective by Diener vs. eudemonistic perspective by Ryff). Moreover, in the introduction there should be more discussed the debate on the process of the adaptation of subjective well-being (SWB) as result of experiencing various stressful life events (see stress faced by the students, including ordinary daily stress, but also traumatic stress). The authors should indicate to what extent well-being is stable (see hedonic treadmill” model; Lyubomirsky, 2011), and also in-born, personality oriented feature (see the role of personality traits in well-being and its change) or is changeable and depend from the current life-situations and stressors. What is the difference in adaptation of well-being in general versus clinical samples. Thus, please discuss and cite following papers on that issue:
Diener, E., Heintzelman, S. J., Kushlev, K., Tay, L., Wirtz, D., Lutes, L. D., & Oishi, S. (2016). Findings All Psychologists Should Know From the New Science on Subjective Well-Being. Canadian Psychology/Psychologie Canadienne, doi:10.1037/cap0000063
Lyubomirsky, S., Sheldon, K. M., & Schkade, D. (2005). Pursuing happiness: The architecture of sustainable change. Review of General Psychology, 9, 111–131.
Ryff, C. (2014). Psychological Well-Being Revisited: Advances in the Science and Practice ofnEudaimonia. Psychotherapy and Psychosomatics, 83, 10–28. doi: 10.1159/000353263
Rzeszutek, M., Gruszczyńska, E., Firląg-Burkacka, E. (2018) Socio-medical and personality correlates of psychological well-being among people living with HIV: a latent profile analysis. Applied Research in Quality of Life, 14, 1113-1127. doi: 10.1007/s11482-018-9640-1.1037/1089-2680.9.2.111
Response: We appreciate your comments and suggestions. We have revised our manuscript accordingly.
Finally, at the end of introduction, before the method section please provide current study section, which you highlight aims of the study and hypotheses.
Response: We have revised our manuscript accordingly.
Method – this part is described well, but please put the table with sociodemographic information of the sample. Did participants receive some renumeration for study participation?
Response: We have revised our manuscript as suggested. The participants did not receive any remuneration.
Results – this part is very well presented.
Discussion – this part, like introduction, is poorly grounded in the classical literature on well-being. Thus, I recommend also to elaborate on this issue more explaining the results they obtained - above papers to cite and discuss also here will be helpful.
Taking everything into an account, I recommend publishing this manuscript in the Healthcare. However, it needs before some major revisions, which I highlighted above.
Response: We thank you again for your positive and useful suggestions.
Round 2
Reviewer 1 Report
Thank you for addressing the recommendations made by the reviewers.
I have only one question. The question on sexual orientation (sociodemographic information) includes the LGBTQ category. What is the difference between Homosexual and this category? Homosexual people, couldn't you answer both categories?
Author Response
Dear reviewer,
Thank you for pointing this out. You are right. We have corrected this point.
It now reads,
Homosexual
Heterosexual
Bisexual
Others

Reviewer 3 Report
The authors addressed my comments well. I suggest publish this article in the present form.
Author Response
Dear reviewer,
Thank you very much for your acceptance.
Best Regards,
TW